# Relationship between Liver Stiffness and Steatosis in Obesity Conditions: In Vivo and In Vitro Studies

**DOI:** 10.3390/biom12050733

**Published:** 2022-05-23

**Authors:** Francesca Baldini, Mohamad Khalil, Alice Bartolozzi, Massimo Vassalli, Agostino Di Ciaula, Piero Portincasa, Laura Vergani

**Affiliations:** 1Department of Earth, Environment and Life Sciences (DISTAV), University of Genova, Corso Europa 26, 16132 Genova, Italy; baldinifrancesca92@gmail.com; 2Nanoscopy, Istituto Italiano Tecnologia, Via Enrico Melen 83, 16152 Genova, Italy; 3Clinica Medica “A. Murri”, Department of Biomedical Sciences and Human Oncology, University of Bari Medical School, Piazza Giulio Cesare 11, 70124 Bari, Italy; mak_37_47@hotmail.com (M.K.); agostinodiciaula@tiscali.it (A.D.C.); piero.portincasa@uniba.it (P.P.); 4Dipartimento di Ingegneria dell’Informazione, Università degli Studi di Firenze, Via di S. Marta 3, 50139 Firenze, Italy; alice.bartolozzi91@gmail.com; 5James Watt School of Engineering, University of Glasgow, Glasgow G12 8LT, UK; massimo.vassalli@glasgow.ac.uk

**Keywords:** obesity, non-alcoholic fatty liver disease (NAFLD), hepatocyte steatosis, liver stiffness, single cell biomechanics, ultrasonography, elastography

## Abstract

Obesity is a major risk factor for metabolic dysfunction such as non-alcoholic fatty liver disease (NAFLD). The NAFLD spectrum ranges from simple steatosis, to steatohepatitis, fibrosis, and cirrhosis. The aim of this study is to characterize the grade of steatosis being associated with overnutrition and obesity, both at the level of single hepatocyte and whole liver, and to correlate it with the hepatocyte/liver stiffness and dysfunction. For the in vivo study, 60 subjects were enrolled and grouped based on the stage of liver steatosis/fibrosis according to biochemical analyses, liver ultrasonography (USG) and acoustic radiation force impulse shear wave elastography (ARFI-SWE). For single hepatocyte analyses we employed in vitro models of moderate and severe steatosis on which to assess the single cell biomechanics by Single Cell Force Spectroscopy (SCFS) and Quantitative Phase Microscopy (QPM). Results show that in vivo liver stiffness depends mainly on the extent of fat accumulation and not on fibrosis. These results parallel the in vitro observations showing that hepatocyte stiffness and dysfunction increase with increasing fat accumulation and lipid droplet enlargement. Our findings indicate that the extent of steatosis markedly affects the biomechanical properties of both liver and single hepatocytes thus proving insights about the role of modulation of liver/hepatocyte elasticity as a physical mechanism transducing the obesity-dependent excess of plasmatic lipids towards liver steatosis and dysfunction.

## 1. Introduction

The first consequence of overnutrition is excessive fat accumulation in the adipose tissue leading to overweight and obesity, both major public health problems worldwide [1,2]. Obesity impacts the liver through the release of a panel of adipokines and hormones [3,4], which are balanced in physiological conditions, but this balance is disrupted in obesity. At the molecular level, a continuous and dynamic antagonism links in vivo adipokines/cytokines with a favorable effect to ones with unfavorable impact on the liver, with the former defending, and the latter promoting steatosis, inflammation, and fibrosis [5].

Diets high in fat and sugar promote the development of obesity, and recent data report that saturated fats have lost the position of the main steatogenic and damaging foods, in favor of excess carbohydrates. In particular, high-intake of fructose is highly steatogenic, and, as fructose consumption has increased worldwide, is resulting in metabolic disorders such as obesity, insulin resistance, and diabetes [6,7,8].

The steady state concentration of hepatic triglycerides (TGs) and fatty acids (FAs) is low under physiological conditions, as the liver is not a primary fat depot. Over-accumulation of TGs in hepatocytes (≥5% of liver parenchima) leads to nonalcoholic fatty liver disease (NAFLD), a spreading chronic liver disease worldwide as a consequence of overnutrition [9,10,11,12]. NAFLD may encompass a spectrum of liver abnormalities ranging from simple steatosis, to nonalcoholic steatohepatitis (NASH), cirrhosis, and hepatocarcinoma [13] with important economic and clinical burden [14,15,16]. Recently, the re-definition of NAFLD to metabolic (dysfunction)-associated fatty liver disease (MAFLD) has been proposed to emphasize the association of liver steatosis with metabolic dysregulation [17,18], such as overweight/obesity, and/or type 2 diabetes mellitus (T2DM), or other metabolic disorders [19,20].

In the cell, TGs are stored in the form of cytosolic lipid droplets (LDs) [21,22], which are composed of a neutral lipid core surrounded by phospholipids and LD-associated proteins controlling lipid metabolism and traffic [23]. Among them, the adipose differentiation-related protein (ADRP) regulates formation and structural maintenance of LDs and its overexpression stimulates lipogenesis and inhibits lipolysis, thus functioning as a marker for the extent of lipid accumulation [24]. ADRP gene expression is under the control of the peroxisome proliferator-activated receptors (PPARs), which are a family of ligand-dependent transcription factors involved in NAFLD progression [25,26]. At the molecular level, the progression of NAFLD is typically associated to increased apoptosis, oxidative stress, and up-regulation of IκB kinase β-interacting protein (IkBip) [27].

In NAFLD patients, steatosis typically appears as macrovesicular steatosis (large LDs), which has rather good long-term prognosis and rare progression to fibrosis or cirrhosis [28]. However, at the cellular level, excess accumulation and enlargement of LDs may displace the nucleus to cell periphery and modify cell biomechanics [29]. Moreover, as LDs are stiffer than the aqueous cytosol, their excess may mechanically distort the intracellular environment [29]. Indeed, the onset and progression of NAFLD in vivo is associated with an altered mechanical liver phenotype, in which the stiffness is strictly linked to organ dysfunction and used as a diagnostic marker [30]. In vivo, liver stiffness demonstrated considerable capability in identifying the stages of liver fibrosis better than the serum biomarkers [31]. Although, liver biopsy is still regarded as the standard of reference for liver fibrosis assessment, the procedure is invasive, so a combination of serum biomarkers and liver stiffness values is an approach of increasing interest [32,33].

Despite recent findings, the connection between biomechanical properties and physiological functions of both the single hepatocyte and the whole liver has not been yet identified, and requires further investigation. In the present study, we investigated the biomechanical properties of the liver as a consequence of steatosis both at the level of the whole organ and isolated hepatocytes. Liver stiffness in adult subjects with a different grade of steatosis was assessed by Abdominal Ultrasonography (US) and Acoustic Radiation Force Impulse Shear Wave Elastography (ARFI-SWE), two standard techniques for liver imaging [34]. On the same subjects the serologic tests were also performed. The mechanical properties of cultured hepatocytes being associated to the grade of steatosis were assessed at the single cell level in different steatotic models mimicking both moderate and severe steatosis by means of Single Cell Force Spectroscopy (SCFS) [35] and high resolution Quantitative Phase Microscopy (QPM) [36].

## 2. Materials and Methods

### 2.1. Chemicals

All chemicals, unless otherwise indicated, were supplied by Sigma-Aldrich Corp. (Milan, Italy).

### 2.2. In Vivo Study

#### 2.2.1. Subject Recruitment

Patients were enrolled on an outpatient basis at the Division of Internal Medicine, Regional Hospital “Policlinico”, Bari, Italy. Subjects entering the study gave full written informed consent and underwent clinical assessment by history and full physical exam. Chronic liver disease other than NAFLD was excluded by accurate clinical assessment. In the study we included 20 consecutive patients with moderate or severe NAFLD, and 40 subjects with normal liver or mild steatosis as assessed by ultrasound (see below). We decided to consider as a unique group (Controls) subjects with ultrasonographic findings of normal liver and mild steatosis, since the accuracy of ultrasound is poor in detecting a fat accumulation in the liver <30%, whereas the presence of posterior attenuation and/or skip areas are closely related to steatosis of > or = 30% [37]. To avoid misclassifications due to inter-operator variability [38], all examinations were performed by a single, experienced operator. In all subjects the following anthropometric variables were assessed: body weight (kg), height (cm), and body mass index (BMI), calculated as the Quetelet’s index (kg/m^2^). BMI ranging from 18.5 to 24.9 kg/m^2^ defined normal weight subjects, while a BMI ranging from 25.0 to 29.9 kg/m^2^ and >30 kg/m^2^ defined overweight and obese subjects, respectively. The protocol was approved by the local Ethics Committee (study number 5408, protocol number 0013869; AOUCPG23/COMET/P). Before the study, all subjects gave full written informed consent to allow all authors to access and use the data for research purposes.

#### 2.2.2. Serum Biomarker Panels

The serum biomarkers APRI and FIB-4 were calculated from AST (in units per liter), platelet count (in 10^9^ cells per liter), ALT (in units per liter), and age (in years) by using the following two formulas: APRI = (AST/upper limit of normal)/platelet count × 100 and FIB-4 = (age × AST)/(platelet count × ALT1/2). The upper limit of normal for both AST and ALT was 40 U/L. An APRI score greater than 0.7 had a sensitivity of 77% and specificity of 72% for predicting significant hepatic fibrosis [39]. For FIB-4, a FIB-4 score >2.67 has a 97% specificity in diagnosing advanced fibrosis, whereas a score <1.45 can exclude fibrosis (negative predictive value of 90%) [40].

#### 2.2.3. Liver Steatosis and Stiffness Measurements

Abdominal ultrasonography (USG) and acoustic radiation force impulse shear wave elastography (ARFI-SWE) were performed using a 6C1 (6 MHz) abdominal curved array transducer. Liver steatosis was measured with the ‘Hitachi Noblus-E echocolordoppler’ (Hitachi Medical, Tokyo, Japan) and Logiq E9 (GE, Healthcare) ultrasound equipment, using a 3.5 MHz convex probe. Kidney cortex echogenicity (control) was compared with the echogenicity of the liver parenchyma. Ultrasonography reliably detects a hyperechoic texture or a bright liver upon diffuse fatty infiltration [41], and represents a non-invasive imaging technique for grading liver steatosis (ranging from grade 0 = absent to grade 3 = severe steatosis) [42,43,44,45,46]. Three groups of patients were identified depending on the grade of the liver steatosis: absent/mild (normal liver echogenicity or isolate finding of liver echogenicity brighter than the renal cortex), moderate (liver echogenicity brighter than the renal cortex, associated with hepatic and/or portal venous margin blurring), and severe (additional presence of diaphragmatic attenuation from liver fat). By ARFI-SWE, a total of 10 valid measurements were obtained in the right lobe of the liver by placing Regions of Interest (ROIs) in the liver tissue at a minimum distance of 1 cm from the liver capsule, and in a liver area free of large blood vessels, bile ducts, and artifacts. The patient was asked to hold their breath at the time of measurement, and the axis of ROI was placed perpendicular to the capsule of the liver. The median of 10 measurements was thereafter considered as the final shear wave velocity (m/s) of the liver, a value directly proportional, to local tissue stiffness [47].

### 2.3. Cell Culture and Treatments

Rat hepatoma FaO cells were obtained from European Collection of Authenticated Cell Cultures (ECACC- Salisbury, Wiltshire, UK). Cells were cultured in low glucose (1.8 g/L) Coon’s modified Ham’s F12 supplemented with 2 mM Glutamine and 10% Foetal Bovine Serum (FBS) (Euroclone Milan, Italy) at 37 °C in a humidified atmosphere containing 5% CO_2_ [48]. For treatments, FaO cells were grown on plastic dishes in high-glucose (4.5 g/L) medium with 0.25% bovine serum albumin (BSA), then subjected to different treatments. Incubation with oleate/palmitate mixture (2:1 molar ratio, final concentration 0.75 mM) for 3 h resulted in a condition of moderate steatosis (MSt), while incubation with fructose 5.5 mM for 72 h and sequential incubation with oleate/palmitate mix for 3 h led to a condition of severe steatosis (SSt). No treatment was applied for control cells (Ctrl).

### 2.4. Lipid Droplet Imaging

Cells were grown on collagen-coated glass slides, washed with PBS and fixed in 4% formaldehyde/PBS for 20 min. Neutral lipids of LDs were visualized using the selective Oil Red O (ORO) dye [49]. Cells were stained for 30 min with 0.3% ORO solution which was freshly prepared from a stock solution of 0.5% in isopropanol, washed, and mounted. LDs were also visualized using the fluorescent probe BODIPY 493/503 (Molecular Probes, Life Technologies, Monza, Italy) [27]. After fixation and washing, cells were incubated with 1 µg/mL BODIPY 493/503 in PBS for 30 min, washed, and mounted with 4′,6-diamidino-2-phenylindole (DAPI). Images were obtained using a Leica DMRB light microscope equipped with a Leica CCD camera DFC420C (Leica, Wetzlar, Germany). In each population, more than 500 cells were analyzed.

### 2.5. RNA Extraction and Real-Time qPCR

RNA was isolated using Trizol reagent, cDNA was synthesized and quantitative real-time PCR (qPCR) performed in quadruplicate using 1× IQ^TM^SybrGreen SuperMix and Chromo4^TM^ System apparatus (Biorad, Milan, Italy). The relative quantity of target mRNA was calculated by the comparative Cq method using glyceraldehyde 3-phosphate dehydrogenase (GAPDH) as housekeeping gene, and expressed as fold induction with respect to controls. Primer pairs designed ad hoc starting from the coding sequences of *Rattus norvegicus* (http://www.ncbi.nlm.nih.gov/Genbank/GenbankSearch.html, accessed on 22 May 2022) and synthesized by TibMolBiol (Genova, Italy) are reported in previous papers [27].

### 2.6. Single Cell Force Spectroscopy (SCFS)

Single cell mechanical measurements were obtained using a nano-indentation device (Piuma Chiaro, Optics11, Amsterdam, NL, USA) mounted on an inverted microscope. Living cells were measured in Petri dishes in the culture medium and indented using a glass cantilever wit a spherical tip (radius of 9 μm and stiffness of 0.2 N/m). The elastic parameters were calculated based on the Hertzian dynamics and the curves were analyzed following an established protocol [http://doi.org/10.3791/63401, accessed on 22 May 2022]. For each experimental condition, about 50–70 curves were acquired over at least three different repeats after calibration.

### 2.7. Quantitative Phase Microscopy (QPM)

*QPM* aims to reconstruct the three-dimensional (3D) shape of semi-transparent objects by recording the phase of the light. Cells grown on coverslips were fixed in 4% paraformaldehyde for 20 min and washed in PBS; then slides were mounted and observed. A home-made acquisition system equipped with a Nikon 40× infinity-corrected objective (NA 0.75; WD 0.66 mm), mounted on a motorized Z-axis with a 0.01 μm resolution step, and standard modular components (Optem FUSION, Qioptiq Photonics GmbH & Co KG, Goettingen, Germany)) was employed. An area of about 1 cm^2^ was imaged, resulting in 80–100 stack per sample, each comprising 15 Z planes acquired with a 1μm step around the estimated best focal plane. Each Z-stack was transformed in a 3D phase map and the reconstruction was performed using a procedure previously described [50]. A set of morphometric indicators was calculated for each cell, including: height of the cells (H_C_), surface extension (S_E_), and cell contact area (A_C_).

### 2.8. Statistical Analysis

Data are expressed as means ± SD of at least three independent biological experiments performed as technical triplicates. Results from in vivo studies are expressed as mean ± SEM or as proportion (categorical variables). Statistical analysis was performed using *ANOVA* with Tukey’s post-test (biological experiments) or Fisher’s LSD Multiple-Comparison test (in vivo studies) (GraphPad Software, Inc., San Diego, CA, USA and SigmaStat software, https://systatsoftware.com, accessed on 22 May 2022). Multivariate linear regression analysis was used after testing of normal Gaussian distribution and logarithmic transformation. Proportions were compared by the chi-squared test. Statistical significance was assumed for *p*-values ≤ 0.05.

## 3. Results

### 3.1. In Vivo Liver Stiffness and Serum Biomarkers Are Associated to Different Steatosis Stages

Enrolled subjects were classified into three groups according to the grade of liver steatosis, assessed by USG. The three subgroups were homogeneous for age (mean age of 52.8 ± 1.3 years) and gender distribution. The control group (Ctrl) included 40 subjects with normal liver/mild steatosis (US value ≤ 1). The group with moderate liver steatosis (MLSt) included 11 subjects (US value = 2), whereas 9 subjects showed severe liver steatosis (SLSt) (US value = 3) (Table 1).

While the BMI progressively increased across these three subgroups, no significant difference was noticed in serum indices, except serum triglycerides that increased significantly in subjects with moderate steatosis as compared to controls (213 ± 27.8 mg/dL in MLSt vs. 105.3 ± 13.0 g/dL in Ctrl). Also, APRI and FIB-4 values were similar among the three groups (Table 1).

Regarding obesity, it was more frequent in subjects with moderate or severe steatosis with respect to controls. When liver stiffness was measured by ARFI, the values progressively increased from 1.32 ± 0.04 m/s in controls (Ctrl) to 1.52 ± 0.08 m/s and 1.58 ± 0.09 m/s in subjects with moderate (MLSt) and severe liver steatosis (SLSt), respectively (*p* < 0.01) (Table 1 and Figure 1A). ARFI showed also a normal/mild (i.e., F1–F2) grade of fibrosis for most enrolled subjects (93%). The absence of a significant liver fibrosis was confirmed by both APRI values (average < 0.7 in all subgroups) and FIB-4 values (average < 0.45 in all subgroups) (Table 1). Lastly, a positive correlation (*p* < 0.001) was showed between the BMI and the liver stiffness value when all subjects were considered (Figure 1B).

### 3.2. In Vitro Hepatocyte Stiffness and Biomarker Expression Are Associated to Different Grade of Steatosis

The treatments of hepatocytes with FAs alone or in combination with fructose allowed to mimic in vitro the hepatic steatosis progression from moderate (FAs alone) to severe (FAs combined with fructose) [51]. The spectrophotometric quantification of TG content in hepatocytes showed that, compared to controls, lipid accumulation increased of +57% in the moderate steatosis (MSt) model, and the increase was larger (+277%) in the severe steatosis (SSt) model (*p* ≤ 0.05) (Figure 2A). The microscopic analysis of ORO- and BODIPY-stained hepatocytes showed a marked accumulation of LDs in all steatotic conditions (Figure 2D). While control cells showed only small (about 0.9 ± 0.3 µm) and few (about 2 LDs/cell) cytosolic droplets, the LD number increased similarly in both steatotic conditions (about 13 LDs/cell in MSt cells, and 14 LDs/cell in SSt cells) (Figure 2C). On the other hand, the LD size increased markedly in SSt cells (+334%), and less in MSt cells (+214%) compared to control (Figure 2B).

Hepatic lipid accumulation depends on the fine regulation of lipogenic pathways which are under the master control of PPARγ [52]. The PPARγ mRNA expression was up-regulated in both MSt and SSt models (2.2- and 1.5-fold induction vs. control, respectively; *p* ≤ 0.05) (Figure 3). Similarly, ADRP mRNA expression, a marker for LD accumulation, was significantly increased in both MSt and SSt models (1.85- and 2.08-fold induction vs. control, respectively; *p* ≤ 0.05) (Figure 3). By contrast, IkBip mRNA expression, a marker for hepatic cell damage, was significantly up-regulated only in SSt model (1.61-fold induction vs. control; *p* ≤ 0.05) (Figure 3).

### 3.3. Single Cell Elasticity and Morphometry

The elasticity of living hepatocytes in different steatotic conditions was assessed by SCFS while recording force vs. displacement curves. The relative elasticity Er increased significantly (*p* ≤ 0.05) in MSt model (1.15-fold increase vs. control), but a larger increase occurred in SSt model (1.17- fold increase vs. Ctrl) (Figure 4A) and this clearly indicates that stiffness increases as a function of extent of fat accumulation. In the same cell models, the 3D shape of hepatocytes was reconstructed using QPM which allowed to measure the physical thickness of the sample, point by point, and to quantify some important geometrical at the single cell level. Figure 4 summarizes three important geometrical parameters of cells in different steatotic conditions: cell height (HC), surface extension (SE), and cell contact area (AC). The surface extension was not affected by fat accumulation as it did not significantly change in both MSt and SSt conditions with respect to control (Figure 4B). However, both MSt and SSt resulted to be significantly thinner and larger than controls as indicated by the reduction in HC (−60% and −31% vs. control, *p* ≤ 0.05, respectively) and increase in AC (+50 and +55% vs. control, *p* ≤ 0.05, respectively) (Figure 4C,D).

## 4. Discussion

The widespread epidemic of obesity is directly linked to increasing prevalence of NAFLD which has emerged as the most common chronic liver disease worldwide. At the cellular level, excess lipids accumulate in cytosolic droplets that are stiffer than the aqueous cytosol and might mechanically distort the cell and alter its elasticity [29]. In vivo, liver stiffness measurement is currently employed to determine the extent of liver steatosis and fibrosis. The present findings combining in vitro and in vivo evidence clearly indicate that an increase in liver/hepatocyte stiffness occurs as a direct consequence of the extent of steatosis and of the enlarging of lipid droplets, and, in turn, is associated with body/cell dysfunction.

Overweight and obesity are typical consequence of excess food intake leading to hypertrophic adipose tissues and visceral fats, the first hallmark of metabolic syndrome. In this condition, the liver rapidly becomes a target for the excess circulating FAs and TGs released from adipose tissue, and it develops into a condition of hepatic steatosis whose severity may progress along with NAFLD pathophysiology.

Stiffness in physics defines the extent to which an elastic object resists deformation in response to an applied force. The liver is a viscoelastic structure whose stiffness is affected by diet, inflammation, steatosis, cholestasis, and other pathological factors [53]. In patients with NAFLD, an increase in liver stiffness might precede the development of fibrosis, so the measurement of the liver viscoelastic properties by noninvasive techniques are of important diagnostic value.

The results of the in vivo study reveal an increase in liver stiffness as a direct consequence of hepatic steatosis, rather than secondary to the onset of fibrosis. Indeed, passing from controls to NAFLD patients we observed a progressive increase in liver stiffness with the lowest stiffness value in controls, and the highest value in patients with severe liver steatosis (SLSt). A significant presence of liver fibrosis in NAFLD patients was excluded through noninvasive, alternative methods [34,54]. These methods include the APRI and FIB-4 tests, based on serum biomarkers and age of the subjects [34,54,55], and the ARFI-SWE elastography. The methods supplied mild values of shear wave velocity, FIB-4 values < 0.45, and APRI values < 0.7 in all NAFLD patients thus excluding significant liver fibrosis. Although increased liver stiffness can also derive from portal hypertension [56,57] and liver congestion [58,59], we excluded this cause since patients were accurately assessed at entry, excluding liver disease other than NAFLD (including portal hypertension and liver congestion).

The development of cellular models to mimic in vitro what is occurring in vivo during NAFLD pathogenicity has become a trend in translation medicine. In this regard, our research group developed well-established cellular models for simple, moderate, and severe hepatic steatosis. In particular, the combination of FAs and fructose mimicked a severe steatosis (SSt) with respect to FAs alone leading to a moderate steatosis (MSt). In this way, we paralleled the three groups studied in vivo at the level of isolated hepatocytes. To depict the molecular dysfunction in the hepaotcyte, we assessed the expression of different steatosis markers. Both ADRP and PPARγ expression, which are master regulators of the lipogenic pathways, was up-regulated in MSt and maximally in SSt models. On the other hand, expression of IKbip, a marker for liver damage progression, was unaltered in MSt but it was significantly up-regulated in SSt.

Then, how moderate and severe steatosis can affect single cell mechanical properties was assessed by means of SCFS and QPM. At single cell level, the SCFS data show that the stiffness was increased in MSt cells, but the largest increase was observed in SSt condition. This suggests that the single cell elasticity strictly depends on the steatosis grade and LD enlargement. On the other hand, also the single cell morphometry assessed by QPM was modified as a function of the steatosis grade. Both moderate (MSt) and severe (SSt) steatotic cells appeared larger and thinner than control cells, but they did not change their surface extension. 

Nevertheless data from in vivo examinations are limited by the low number of subjects included in the study, they are in agreement with results from a recent study, showing that the severity of steatosis assessed by a histological score did not affect the association between liver stiffness measurement by elastography and fibrosis in patients with NAFLD [60]. Our results are also in line with previous observations in NAFLD patients explored by transient elastography and histology, showing significantly higher liver stiffness measurements in subjects with low amount of fibrosis and severe steatosis [61]. Interestingly, also a study assessing liver viscoelasticity in an animal model by elastography confirmed a significant correlation between liver elasticity and liver steatosis. We wish to emphasize that the use of liver histology to diagnose advanced fibrosis is actually not accepted [34], due to technical limitations as sampling variability [62] and inadequate inter- hepato-pathologist variability in examining the specimen [63]. By contrast, conventional liver ultrasound is recommended by current guidelines as a first-line tool for the diagnosis of steatosis [55]. Since the main limitation of ultrasound is its poor accuracy in detecting slight (steatosis below 12.5–20%) [64], we considered as a unique group (controls) subjects with US findings of normal liver or mild steatosis, and as NAFLD groups only subjects with clear, objective ultrasonographic signs of fatty liver.

These variations in the biomechanical properties of single hepatocytes could have promising translational applications, especially when we compared with clinical outcomes of in vivo liver stiffness measurements. In this respect, the translational value of the present study derives from the in vivo confirmation of in vitro findings, excluding the possible confounding role of fibrosis and other conditions (i.e., portal hypertension [56,57] and liver congestion [58,59]) able to affect liver stiffness in humans. Despite the previously discussed limitations, the present study offers a comprehensive view of the effect of fat accumulation alone on the biomechanical properties of the liver, demonstrating that liver stiffness increases as a function of the extent of fat accumulation, independently from possible pathologic confounders.

## 5. Conclusions

Obesity, alcohol abuse, genetic disorders, drugs, cholestasis, metabolic disorders, chronic viral hepatitis, and other cryptogenic causes are major factors that promote liver steatosis and fibrosis. Our integrated approach using in vivo and in vitro studies indicates that the intracellular accumulation of fat is able to significantly alter the mechanical properties of the liver and of the single hepatocytes, independently from the presence of fibrosis. Therefore, cell biomechanics may represent a pivotal transducer connecting fat accumulation in cytosolic droplets and cell dysfunction observed in terms of molecule expression. There is hope that improved understanding of the cellular, molecular, and biophysical pathways sustaining the hepatocyte dysfunction due to excess fat accumulation, as that occurring in overnutrition and obesity conditions, might uncover potential therapeutic targets.

## Figures and Tables

**Figure 1 biomolecules-12-00733-f001:**
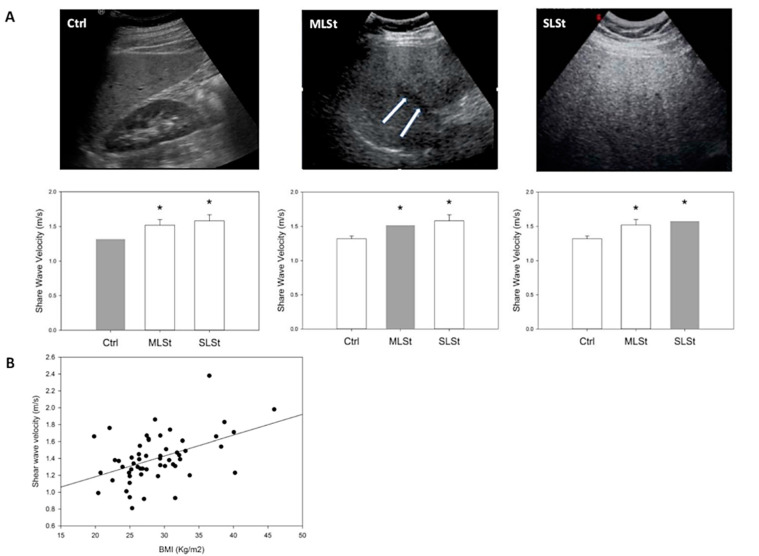
**In vivo liver stiffness and serum biomarkers measurements.** (**A**) Ultrasonographic grading of liver steatosis (upper figures) and corresponding average shear wave velocity (lower graphs, red bars). Ctrl (absent/mild steatosis): a normal liver echogenicity or isolate finding of liver echogenicity brighter than the renal cortex; MLSt (moderate steatosis): liver echogenicity brighter than the renal cortex, associated with hepatic and/or portal venous margin blurring (arrows); SLSt (severe steatosis): additional presence of diaphragmatic ultrasound wave attenuation. Asterisks indicate significantly higher average shear wave velocity values in moderate and severe steatosis, as compared with normal/mild steatosis (ANOVA followed by Fisher’s LSD Multiple-Comparison test). (**B**) Linear regression analysis between Body Mass Index (Kg/m^2^) and acoustic radiation force impulse shear wave velocity (a value directly proportional to local tissue stiffness), measured in a group of 60 adults. R = 0.46, *p* = 0.0003.

**Figure 2 biomolecules-12-00733-f002:**
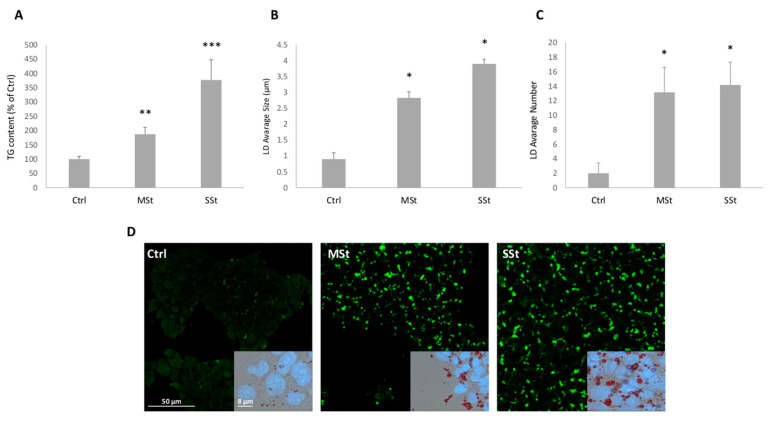
**Lipid accumulation in moderate and severe steatosis models.** For FaO cells incubated in the absence (Ctrl) or in the presence of moderate in vitro steatosis (MSt), and severe steatosis (SSt) we show: (**A**) TG content expressed as percent TG content relative to controls, normalized for proteins determined with Bradford assay. (**B**,**C**) Average size of LDs. and number of LDs/cell. Values are mean ± S.D. from at least three independent experiments. Statistical significance between groups was assessed by ANOVA followed by Tukey’s test. Symbols: Ctrl vs. all treatments * *p* ≤ 0.05. (**D**) Microphotographs of cells stained with BODIPI 493/503 (magnification 20×; Bar: 50µm) and microphotographs of cells stained simultaneously with ORO and DAPI (magnification 40×; Bar: 8µm) were captured. For microscopy analyses a Leica DMRB light microscope equipped with a Leica CCD camera DFC420C was employed. Symbols: Ctrl vs. all treatments * *p* ≤ 0.05, ** *p* ≤ 0.01, *** *p* ≤ 0.001.

**Figure 3 biomolecules-12-00733-f003:**
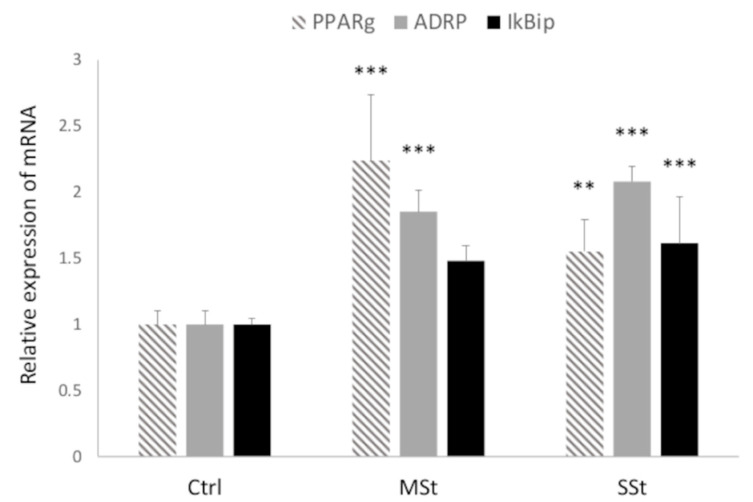
**Modulation of cell function in moderate and severe steatosis models.** The mRNA expression of PPARγ, ADRP and of IkBip were evaluated by qPCR using GAPDH as the internal control. Data are expressed as fold induction with respect to controls. Bars represent SD. ANOVA followed by Tukey’s test was used to assess the statistical significance between groups. Symbols: Ctrl vs. all treatments ** *p* ≤ 0.01, *** *p* ≤ 0.001.

**Figure 4 biomolecules-12-00733-f004:**
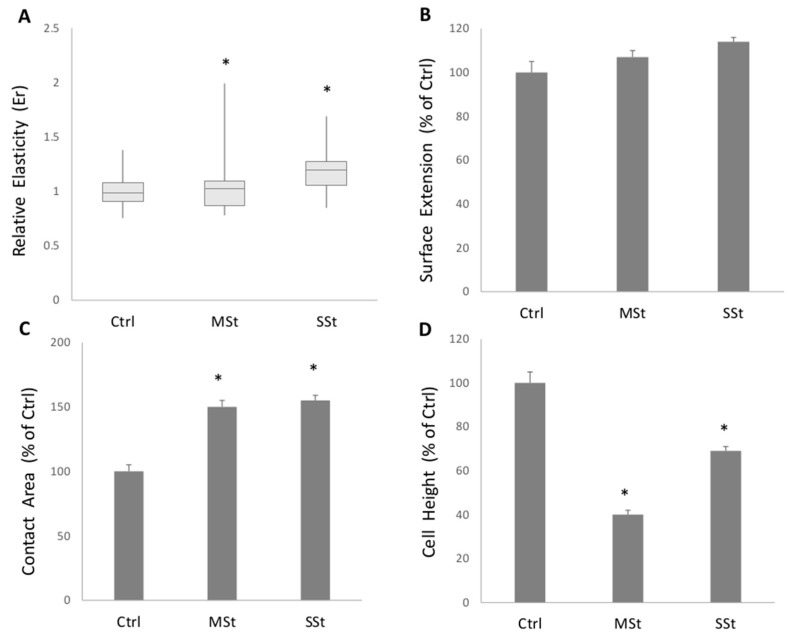
**Changes in single cell biomechanical properties in moderate and severe steatosis models.** (**A**) Relative Elasticity (Er) of single cell respect to the control, obtained through the FIEL method. Symbols: Ctrl vs. all treatments * *p* ≤ 0.05. The morphometric indicators computed for all the treatments are presented relative to the controls: (**B**) surface extension (SE); (**C**) cell contact area (AC); (**D**) height of the cells (HC). Statistical significance respect to the control is marked when at least * *p* ≤ 0.05.

**Table 1 biomolecules-12-00733-t001:** General characteristics, liver stiffness and indices of liver fibrosis in a group of 60 adults selected according to the presence/extent of liver steatosis by USG. *p* value was assessed by ANOVA followed by Fisher’s LSD Multiple-Comparison test. * *p* < 0.05 vs. Ctrl; ^&^ *p* < 0.05 vs. MLSt.

	Normal Liver/Mild Steatosis (Ctrl)	Moderate Liver Steatosis (MLSt)	Severe Liver Steatosis(SLSt)	p (ANOVA)
* **Subjects** *				
Number (%)	40 (66.7%)	11 (18.3%)	9 (15%)	-
Age (years)	51.7 ± 1.6	52.9 ± 3.0	57.9 ± 4.3	0.25
Males n. (%)	20 (50%)	5 (33%)	6 (67%)	-
BMI (Kg/m^2^)	26.7 ± 0.7	32.4 ± 1.4 *	34.3 ± 1.4 *	0.0000005
N. normal weight subjects (%)	14 (35%)	0	0	-
N. of overweightsubjects (%)	20 (50%)	3 (27.3%)	1 (11.1%) *	-
N. of obese subjects (%)	6 (15%)	8 (72.7%) *	8 (88.9%) *	-
* **Serum Biomarkers** *				
Total bilirubin (mg/dL)	0.6 ± 0.05	0.7 ± 0.1	0.7 ± 0.1	0.9
AST (U/L)	22.0 ± 1.7	24.2 ± 3.2	26.8 ± 3.5	0.4
ALT (U/L)	34.1 ± 3.7	42.9 ± 6.9	38.9 ± 7.6	0.5
GGT (U/L)	33.5 ± 6.8	42.8 ± 12.7	54.6 ± 14.0	0.3
Platelets (×10^3^/µL)	224 ± 8	249 ± 16	223 ± 17	0.3
Total cholesterol (mg/dL)	197.7 ± 8.8	202.8 ± 19.0	178.0 ± 14.0	0.4
HDL cholesterol	58.2 ± 4.7	39.8 ± 10	55.8 ± 7.5	0.2
LDL cholesterol	118.5 ± 7.6	124.8 ± 16.3	105.3 ± 12.2	0.5
Triglycerides	105.3 ± 13.0	213 ± 27.8 *	132.1 ± 20.9 ^&^	0.006
* **Grading of Fibrosis** *				
* **ARFI-SWE** *				
Shear wave velocity (m/s)	1.32 ± 0.04	1.52 ± 0.08 *	1.58 ± 0.09 *	0.01
F1 n. (%)	37	7	6	-
F2 n. (%)	2	3	1	-
F3 n. (%)	1	1	1	-
F4 n. (%)	0	0	1	-
* **APRI** *	0.26 ± 0.02	0.26 ± 0.04	0.32 ± 0.04	0.4
* **FIB-4** *	0.92 ± 0.06	0.85 ± 0.12	1.2 ± 0.13	0.12

## Data Availability

Data supporting reported results can be obtained by reasonable request to authors.

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
