# Peer review of "Relationship between Liver Stiffness and Steatosis in Obesity Conditions: In Vivo and In Vitro Studies"

_biomolecules, 2022, doi:10.3390/biom12050733_

Round 1

Reviewer 1 Report

I reviewed with great interest the paper by Baldini F and colleagues, who evaluated the relationship between hepatic rigidity and the presence of steatosis in patients with NAFLD in an elegant mixed basic and clinical study. The authors, based on their results in vivo (60 patients stratified between controls, moderate and severe steatosis) and in vitro in analyzes of single hepatocytes by SCFS and QPM, concluded that hepatic stiffness mainly depended on the extent of accumulation of fat and not fibrosis and was determined primarily by the stiffness of the hepatocytes via the accumulation of fat and the enlargement of lipid droplets. The paper is well written, sound scientific and original. No ethical concerns raised. However, there are few issues would be improved.

  • The authors stated: "Of note, these results were not linked to the presence of liver fibrosis, since both ultrasonographic measurement (i.e., share wave velocity) and serum biomarkers (i.e., APRI ad FIB-4) excluded a significant liver fibrosis.” However, much evidence has shown a direct correlation between LSM, FIB4, and liver fibrosis (1111/liv.14799); I think this statement should be mitigated and better discussed.
  • Relationship between liver stiffness, portal hypertension degree, and congestion have been extensively reported. Authors should consider this and try to link these in vivo experiences with their results.

Author Response

We thank the reviewer for his/her positive comments to our work. We are grateful for the suggestions that allowed us to clarify some points and improve the manuscript.

Moving to the specific comments, we are pleased to provide a point-to-point reply:

  • The authors stated: "Of note, these results were not linked to the presence of liver fibrosis, since both ultrasonographic measurement (i.e., share wave velocity) and serum biomarkers (i.e., APRI ad FIB-4) excluded a significant liver fibrosis.” However, much evidence has shown a direct correlation between LSM, FIB4, and liver fibrosis (1111/liv.14799); I think this statement should be mitigated and better discussed. Relationship between liver stiffness, portal hypertension degree, and congestion have been extensively reported. Authors should consider this and try to link these in vivo experiences with their results.

This critical topic has been extensively discussed in the revised version of the manuscript. In particular, the following text has been now added to the manuscript:

Line 363-366: “Although increased liver stiffness can also derive from portal hypertension and liv-er congestion, we excluded this cause since patients were accurately assessed at entry, excluding liver disease other than NAFLD (including portal hypertension and liver congestion).

Reviewer 2 Report

The paper presents an elegant and comprehensive piece of research concerning evaluation of the grade of steatosis of liver cells and whole organ, and its correlation with the hepatocyte/liver stiffness and dysfunction. Nevertheless, some minor issues have to be addressed.

SPECIFIC COMMENTS:

Line 26 „liver ultrasonography (AU)” and line 133 „Abdominal ultrasonography (AU)”- I would recommend changing the abbreviation AU to USG, what is more clear for a reader.

Line 138 and 139-„ Ultrasonography reliably detects a hyperechoic texture or a bright liver upon diffuse fatty infiltration [40], and represents a non-invasive marker for liver steatosis”- change to non-invasive imagine technique

Line 384 „ and several cryptogenic causes”- we know nothing about cryptogenic causes, so we cannot say if they are several or numerous. Please modify the sentence.

Conclusion:

Minor revision is required.

Author Response

We thank the reviewer for the positive comments to our work. The text has been modified accordingly to the requests and we provide a point-to-point reply:

  • “Line 26 „liver ultrasonography (AU)” and line 133 „Abdominal ultrasonography (AU)”- I would recommend changing the abbreviation AU to USG, what is more clear for a reader”

According to the suggestion, the abbreviation “AU” has been changed in “USG” throughout the manuscript.

  • “Line 138 and 139-„ Ultrasonography reliably detects a hyperechoic texture or a bright liver upon diffuse fatty infiltration [40], and represents a non-invasive marker for liver steatosis”- change to non-invasive imagine technique”

According to the reviewer’s request, we modified the text.

  • “Line 384 „ and several cryptogenic causes”- we know nothing about cryptogenic causes, so we cannot say if they are several or numerous. Please modify the sentence.”

The reviewer is absolutely right; the sentence was not clear. We removed “several” from the sentence. The sentence has been changed as follows “Obesity, alcohol abuse, genetic disorders, drugs, cholestasis, metabolic disorders, chronic viral hepatitis, and other cryptogenic causes are major factors…”

Reviewer 3 Report

The authors present an in vivo study on liver stiffness by using ARFI-SWE in humans with no, moderate or severe liver steatosis an well as an in vitro study on stiffness of single cell biomechanics by Single Cell Force Spectroscopy and Quantitative Phase Microscopy on rat hepatoma cells. The authors report and enhanced stiffness in vivo related to fat accumulation, and also increased hepatocyte stiffness and dysfunction with increasing fat accumulation.

The authors conclude that in their study, in vivo liver stiffness depends mainly on the extent of fat accumulation and not on fibrosis.

Several issues have to be considered

1. It is well known that liver stiffness as measured by US techniques increases not only due to fibrosis, but can also be related to other factors such as steatosis, inflammation of congestion.
The measurement of liver stiffness is not limited to the stiffness of the hepatocytes, but takes also into account other contituants of the liver. So this part of the study does not give additional information to what is already known.  How can de authors conclude that the increased stiffness is not related to fibrosis when they did not include patients with liver fibrosis? The issue for clinicians dealing with liver diseases is how to interpret liver stiffness values measured by noninvasive techniques when there are confounding factors such as inflammation, congestion or steatosis.

2. A weakness of the in vivo study is the small number of patients (especially with moderate and severe steatosis) and the absence of histological control.

3. The US evaluation of steatosis is subjective. 

The in vivo study has its limitations as previously explained, and does not add sustantially to what is already known on non-invasive liver stiffness measurement.

The in vivo study on elasticity of single cell biomechanics by single cell force spectroscopy and quantitative phase microscopy is interesting and could be published as such. Combining the 2 aspects of the current study is like comparing 'apples with pears. 

Small remark: on several occasion 'share wave velocity' is used instead of 'shear wave'

Author Response

We are grateful to the reviewer whose criticisms allowed us to clarify some points and improve the manuscript.

Moving to the specific comments, we are pleased to provide a point-to-point reply:

  • “It is well known that liver stiffness as measured by US techniques increases not only due to fibrosis but can also be related to other factors such as steatosis, inflammation of congestion. The measurement of liver stiffness is not limited to the stiffness of the hepatocytes, but takes also into account other constituants of the liver. So this part of the study does not give additional information to what is already known. How can de authors conclude that the increased stiffness is not related to fibrosis when they did not include patients with liver fibrosis? The issue for clinicians dealing with liver diseases is how to interpret liver stiffness values measured by noninvasive techniques when there are confounding factors such as inflammation, congestion or steatosis.”

We absolutely agree with the reviewer. We modified the text by discussing more extensively the potential and limits of noninvasive techniques in the assessment of liver fibrosis, and possible alternative causes of increased liver stiffness. We wish to underline that due to the high prevalence of NAFLD, the use of liver histology to diagnose advanced fibrosis in not accepted , also considering that liver biopsy is not considered a perfect reference standard, due to technical limitations as sampling variability and inadequate inter- hepato-pathologist variability in examining the specimen. Thus, noninvasive, alternative methods have been proposed and validated to assess liver fibrosis in NAFLD, as biomarkers and elastography. Scores based on the assessment of AST, ALT, age and platelet count are indicated to rule out significant fibrosis in NAFLD. The fibrosis 4 calculator (FIB-4) has been extensively validated in different populations with NAFLD, and is able to determine the risk of advanced fibrosis in NAFLD patients. In particular, a FIB-4 score >2.67 has a 97% specificity in diagnosing advanced fibrosis, whereas a cutoff value lower than 1.45 can exclude advanced fibrosis (negative predictive value of 90%). Similarly, a significant hepatic fibrosis can be adequately assessed by an APRI score greater than 0.7. On the other hand, 2D shear wave elastography allows a precise non-invasive staging of liver fibrosis in NAFLD patients, with a strong correlation between histological fibrosis stages and 2D shear wave measurements. All these tests point, in the present study, towards the lack of a significant liver fibrosis in NAFLD patients, who showed mild values of shear wave velocity, FIB-4 value lower than 0.45 and APRI value <0.7 in all subgroups. Available evidence indicates that an increased liver stiffness can also derive from portal hypertension and liver congestion. However, this is not the case in the present series, since patients were accurately assessed at entry, and liver disease other than NAFLD (including portal hypertension and liver congestion) were exclusion criteria.

To clarify the above points we modified the text as follows:

Line 358-366: A significant presence of liver fibrosis in NAFLD patients was excluded through non-invasive, alternative methods [34,54]. These methods include the APRI and FIB-4 tests, based on serum biomarkers and age of the subjects [34,54,55], and the ARFI-SWE elas-tography. The methods supplied mild values of shear wave velocity, FIB-4 values <0.45, and APRI values <0.7 in all NAFLD patients thus excluding significant liver fibrosis. Although increased liver stiffness can also derive from portal hypertension [56,57] and liver congestion [58,59], we excluded this cause since patients were accurately assessed at entry, excluding liver disease other than NAFLD (including portal hypertension and liver congestion).

  • “A weakness of the in vivo study is the small number of patients (especially with moderate and severe steatosis) and the absence of histological control.”

We agree with the reviewer about the small number of observations.  This limitation has been now reported and commented in the discussion section.

Regarding the criticism about the absence of histological control we discussed this aspect in the discussion section underlining that due to the high prevalence of NAFLD, the use of liver histology to diagnose advanced fibrosis in not accepted [53], also considering that liver biopsy is not considered a perfect reference standard [54] due to technical limitations as sampling variability [55] and inadequate inter- hepato-pathologist variability in examining the specimen [56]. Thus, noninvasive, alternative methods have been proposed and validated to assess liver fibrosis in NAFLD, as biomarkers and elastography [53,57].

  • “The US evaluation of steatosis is subjective.”

The reviewer is absolutely right and we included a sentence in the Methods section:

Line 114-116: “To avoid misclassifications due to inter-operator variability [38], all examinations were performed by a single, experienced operator.”

  • The in vivo study has its limitations as previously explained, and does not add substantially to what is already known on non-invasive liver stiffness measurement. The in vivo study on elasticity of single cell biomechanics by single cell force spectroscopy and quantitative phase microscopy is interesting and could be published as such. Combining the 2 aspects of the current study is like comparing 'apples with pears. 

The aim of the present study was to focus on the relationships between hepatocyte fat accumulation and mechanical liver phenotype, with a comprehensive in vitro and in vivo point of view. In this respect, the novelty of the present study has been more extensively commented in the Discussion section:

Line 405-412: “In this respect, the translational value of the present study derives from the in vivo con-firmation of in vitro findings, excluding the possible confounding role of fibrosis and other conditions (i.e., portal hypertension [56,57] and liver congestion [58,59]) able to affect liver stiffness in humans. Despite the previously discussed limitations, the pre-sent study offers a comprehensive view of the effect of “pure” fat accumulation on the biomechanical properties of the liver, demonstrating that liver stiffness increases as a function of the extent of fat accumulation, independently from possible pathologic confounders.”

  • “Small remark: on several occasion 'share wave velocity' is used instead of 'shear wave'”

Thanks for this comment. The type mistakes have been now corrected throughout the manuscript.

Round 2

Reviewer 3 Report

The authors have appropriately addressed the reviewers' comments.